# Identification of *Aeromonas veronii* as the Pathogen Associated with Massive Mortality in Bronze Gudgeon (*Coreius heterodon*)

**DOI:** 10.3390/ani14162440

**Published:** 2024-08-22

**Authors:** Wenzhi Liu, Mengmeng Li, Mingyang Xue, Yong Zhou, Nan Jiang, Yan Meng, Yisha Liu, Jingwen Jiang, Xiaolin Liao, Yuding Fan

**Affiliations:** 1Yangtze River Fisheries Research Institute, Chinese Academy of Fishery Sciences, Wuhan 430223, China; liuwenzhialisa@yfi.ac.cn (W.L.); lmm20000312@163.com (M.L.); xmy@yfi.ac.cn (M.X.); zhouy@yfi.ac.cn (Y.Z.); jn851027@yfi.ac.cn (N.J.); mengy@yfi.ac.cn (Y.M.); liuys629@163.com (Y.L.); j178381682@outlook.com (J.J.); 2Graduate School of Chinese Academy of Agricultural Sciences, Beijing 100081, China; 3College of Fisheries and Life Sciences, Shanghai Ocean University, Shanghai 201306, China; 4Fisheries College, Hunan Agricultural University, Changsha 410128, China

**Keywords:** bronze gudgeon (*Coreius heterodon*), *Aeromonas veronii*, virulence genes, blood parameters, phylogenetic analysis

## Abstract

**Simple Summary:**

An *Aeromonas veronii* strain, isolated from bronze gudgeon, was identified as a significant cause of morbidity and mortality. Through bacterial isolation, virulence gene analysis, and physiological characterization, we identified this bacterium as containing the act, alt, lip, LuxS, and ascV genes, which are distinct from those reported in infections of other aquatic animals. Designated as strain WH10, this is the first documented instance of *Aeromonas veronii* infecting bronze gudgeon. Our findings provide new insights into the mechanisms of bacterial transmission and underscore the necessity for further research into *Aeromonas veronii* pathogenesis.

**Abstract:**

*Aeromonas veronii*, an opportunistic pathogen toward aquatic organisms, was identified as the causative pathogen (isolate WH10) in diseased bronze gudgeon via bacterial isolation, and morphological, physiological, biochemical, and molecular characterization. WH10 exerted its pathogenicity via five virulence genes, including those encoding cytotoxic enterotoxins (act and alt), lipase (lip), a quorum sensing-controlled virulence factor (LuxS), and a Type III secretion system inner membrane component (ascV). WH10 was shown to be sensitive to compound sulfamethoxazoles, cefothiophene, doxycycline, and sulfamethoxazole. Toward bronze gudgeon, WH10 had a median lethal dose (LD_50_) of 1.36 × 10^6^ colony forming units/mL. Analysis of blood parameters of diseased fish revealed significant increases in monocytes and neutrophils, but decreased numbers of lymphocytes. Serum aspartate aminotransferase activity and triglyceride concentration were significantly higher in diseased fish than in healthy fish. The reverse was noted for alkaline phosphatase, total protein, albumin, total cholesterol, and glucose. Thus, *Aeromonas veronii* is implicated as the causative agent of the mass mortality observed in bronze gudgeon, warranting further investigations into the diagnosis, epidemiology, prevention, and treatment of this infectious disease.

## 1. Introduction

The bronze gudgeon (*Coreius heterodon* Bleeker 1964) is an indigenous fish species in China, classified under the subfamily *Gobioninae* within the order *Cypriniformes*. It has important economic significance among freshwater fish and is the predominant species inhabiting the mid- and upstream regions of the Yangtze River [1,2]. Moreover, the bronze gudgeon is frequently utilized as an indicator species in ecological research to assess water temperature stress and investigate the impacts of reservoir and hydropower operations on fish spawning [3,4,5]. Previous studies on bronze gudgeon have primarily focused on aspects such as population dynamics [1], population genetics [6], the accumulation of metals [7], and body proximate compositions [8]; however, there is a general absence of documented instances of problematic diseases in this species.

However, in October 2023, an outbreak of a severe disease affecting bronze gudgeon occurred in Wuhan city, Hubei province. The diseased fish exhibited notable hemorrhage on their body surface and fins, together with ulcerated skin. The cumulative mortality rate was estimated to exceed 70%, representing the first reported occurrence of such an epidemic in bronze gudgeon. Subsequent investigation suggested that water disinfection or oral administration of antibacterial drugs could effectively control the disease, leading to the suspicion an etiological agent was a bacterium.

The present study aimed to comprehensively investigate an *Aeromonas veronii* [9] pathogen that was isolated from bronze gudgeon in China. We carried out bacterial isolation, bacterial identification, animal experiments, antibiotic susceptibility assays, differential leukocyte counts (DLC), biochemical analysis of serum samples, and screening of virulence-related genes. The isolated *Aeromonas veronii* strain, designated as WH10, was identified to be the bacterium that caused the epidemic affecting bronze gudgeon in Hubei province, China. 

## 2. Materials and Methods

### 2.1. Experimental Bronze Gudgeon

Moribund bronze gudgeon displaying typical clinical signs, with a length ranging from 15 to 42 cm, were sampled from a bronze gudgeon farm in Wuhan city, Hubei province, in October 2023. The diseased bronze gudgeon were promptly preserved in oxygenated bags or on ice before transport to the laboratory for subsequent diagnostic assessment and pathogen identification. 

Healthy bronze gudgeon measuring 15–20 cm in length were sourced from the breeding station of Institute of Hydroecology, MWR & CAS, which did not report any instances of the disease. Before experimental infection, the healthy control fish were allowed to acclimatize in recirculation rearing systems (2 m × 1.5 m × 1.5 m). The water in these systems was aerated and maintained at 25 °C. The fish received a regular commercial diet for 7 days to ensure proper acclimatization.

### 2.2. Histological Analysis

Tissue samples obtained from diseased and healthy bronze gudgeon were initially fixed using 4% paraformaldehyde for 24 h at 4 °C. Subsequently, the samples were washed using Dulbecco’s phosphate-buffered saline (DPBS, Sigma, St. Louis, MO, USA), followed by ethanol gradient (70%, 80%, 90%, 95%, and 100%) dehydration. After dehydration, the samples were subjected to optimum cutting temperature compound embedding. The embedded tissues were then cut into sections using a cryostat (CM1950, Leica, Wetzlar, Germany) at −20 °C. Following sectioning, the samples were stained using Hematoxylin–Eosin and examined under a light microscope equipped with a CCD imaging system (DM2500, Leica, Wetzlar, Germany) [10].

### 2.3. Bacterial Isolation

The diseased fish were sacrificed by employing 0.1% tricaine methane sulfonate (MS-222; Sigma, St. Louis, MO, USA) and sanitized using 75% ethanol before dissection. Tissue samples from the liver and kidney were then streaked separately on Brain Heart Infusion (BHI) agar plates (HopeBio, Qingdao, China) and incubated at 28 °C for 24 h. Following incubation, dominant and uniform bacterial colonies were selected and purified using BHI agar re-streaking. We isolated single colonies, which were preserved in glycerol at −80 °C for subsequent characterization.

### 2.4. Identification of the Bacteria

To characterize the bacterial isolates morphologically, pure colonies were suspended in DPBS, smeared onto glass slides, dried, and Gram stained using a kit (Jiangcheng, Nanjing, China) following the manufacturer’s instructions. An optical microscope (Olympus, Tokyo, Japan) was employed to observe the staining and morphological characteristics. To further analyze the bacteria morphologically, they were fixed using 2.5% glutaraldehyde. Subsequently, the bacteria were subjected to dehydration, drying, and examination under a scanning electron microscope (Hitachi, Tokyo, Japan) [11].

Biochemical tests were conducted utilizing a Biolog microbial identification system (Biolog, Hayward, CA, USA). Pure culture colonies were inoculated into IF-A inoculation solution using a sterile cotton swab. Subsequently, the inoculum of the pure clone was added to GEN III plates (Biolog) at 100 μL per well for analysis using the above-mentioned automatic microbial identification system. Analyses included dextrin, gentiobiose, sucrose, acidic PH PH6, α-D-Lactose, L-Glutamic Acid, Vancomycin, the Voges–Proskauer test (acetoin), raffinose, dulcitol, and fructose. 

Identification of the WH10 strain was further conducted using 16S rRNA gene analysis. WH10 DNA was extracted using a Bacterial Genomic DNA Kit (Tiangen, Beijing, China) according to the supplier’s guidelines. The extracted DNA was employed as the template for PCR amplification. Primer pairs were designed as described previously [12] (Table 1). PCR amplification involved an initial denaturation at 95 °C for 5 min, followed by 35 cycles of denaturation for at 94 °C for 30 s, annealing at 55 °C for 45 s, and extension at 72 °C for 60 s. A final extension step was carried out at 72 °C for 7 min. The resulting amplicons were analyzed using 1.5% agarose gel electrophoresis stained with ethidium bromide for visualization. Following sequencing, the 16S rRNA sequences of the isolate clones were compared with sequences in the NCBI database using BLAST for sequence homology analysis. A phylogenetic tree was constructed using the Maximum-likelihood method in MEGA 7.0 (version 7.0, Mega Limited, Auckland, New Zealand).

### 2.5. WH10 Challenge Experiment

Healthy bronze gudgeon were divided randomly into six groups (*n* = 30 per group) and subjected to a pathogen challenge. The experimental groups comprised bronze gudgeon challenged via intraperitoneal (IP) injection with 0.3 mL of bacterial supernatant at titers of 2 × 10^4.0^, 2 × 10^5.0^, 2 × 10^6.0^, 2 × 10^7.0^, and 2 × 10^8.0^ colony forming units (CFU)/mL, respectively. The control fish received an IP injection of 0.3 mL DPBS. All the fish were reared in tanks containing aerated water at 25 °C for the duration of the challenge assay. Every day, we monitored clinical signs and mortality, and randomly sampled three moribund or dead fish per group (including controls), which were processed for PCR analysis to determine whether the pathogen was present or not. The median lethal dose (LD_50_) of the WH10 strain for bronze gudgeon was determined utilizing the Reed Muench method [17]. Three replicate experiments were performed to ensure the robustness and reliability of the findings.

### 2.6. Susceptibility to Antibiotics Assay

The antibiotic susceptibility of WH10 was evaluated in triplicate using the standard disk diffusion method with compound sulfamethoxazoles, cefothiophene, doxycycline, sulfamethoxazole, gentamicin, amikacin, neomycin, florfenicol, enrofloxacin, and ciprofloxacin. Briefly, WH10 was cultured in BHI medium for 1 day. Thereafter, the WH10 solution was adjusted to 1 × 10^8.0^ CFU/mL with sterile PBS and spread on BHI agar plates. Before the WH10 solution was completely absorbed, antibiotic-containing paper disks (Hangwei, Hangzhou, China) were placed on the agar surface. The plates were then placed in a thermostatic incubator (CIMO, Shanghai, China) for 24 h at 28 °C. The diameter of the inhibition zone around each antibiotic disk was then measured, which was used to determine the sensitivity of WH10 to each antibiotic following the supplier’s guidelines. 

### 2.7. Screening of Virulence-Related Genes

Ten virulence genes were screened using conventional PCR with template DNA extracted from the WH10 strain. These genes included aha, exu, lip, ast, alt, act, LuxS, aerA, hlyA, and ascV. The primers utilized to amplify these genes are detailed in Table 1. Briefly, DNA extracted from WH10 (see Section 2.4) was subjected to PCR as follows: initial denaturation for 5 min at 95 °C; followed by 35 cycles of denaturation for 45 s at 94 °C, annealing for 60 s at 57 °C, extension for 90 s at 72 °C; and a last extension step for 7 min at 72 °C. After sequencing, the virulence-related gene sequences of isolate clones were searched in the NCBI database for sequence homology using BLAST.

### 2.8. Blood Parameters

Leukocytes were counted and classified employing the differential leukocyte counts (DLC) method. Diseased bronze gudgeon (*n* = 3) and healthy bronze gudgeon (*n* = 3) were anesthetized, and a sterile disposable syringe was used to collect blood samples at caudal vertebral vein of fish, which were added immediately to a slide to make a blood smear. From each dish, three blood smears were prepared, followed by Wright–Giemsa staining using the manufacturer’s recommendations (Baso, Zhuhai, China). On each blood smear, 100 white blood cells were classified and counted under the oil lens of a microscope. To quantify serum biochemical markers, 1.5 mL of fresh blood was collected from both diseased and healthy bronze gudgeon. Equal volumes of blood were aliquoted into separate tubes for subsequent analysis. This experiment was conducted in triplicate. The supernatant was obtained by centrifugation at 4000× *g* for 10 min after storage overnight at 4 °C, and then transferred to a new tube for further testing. The serum activity and content of glucose (GLU), total cholesterol (T-CHO), triglyceride (TG), alkaline phosphatase (ALP), alanine aminotransferase (ALT), aspartate aminotransferase (AST), albumin (ALB), and total protein (TP) were assessed using an automatic biochemical analyzer (Sysmex, Kobe, Japan). 

### 2.9. Statistical Analysis

SPSS software was used to analyze the data (version 19.0, IBM Corp., Armonk, NY, USA) and statistical analysis was conducted using analysis of variance (ANOVA). Duncan’s multiple range test was employed to evaluate the significance of the differences between means. Differences were considered significant at *p* < 0.05. In this study, statistical analysis was conducted specifically on the bronze gudgeon blood sample data.

## 3. Results

### 3.1. Clinical Signs of the Disease

In October 2023, a disease outbreak occurred among cultured bronze gudgeon in Wuhan city, Hubei province, China. The disease was highly contagious and resulted in the mortality of bronze gudgeon of all sizes. Clinical manifestations included hemorrhages on the body surface (Figure 1A) and fins (Figure 1C), along with ulcerated skin (Figure 1B). Additionally, diseased fish revealed a dark and enlarged spleen, as well as a hemorrhagic liver (Figure 1A,D). Microscopic examination of the gills from diseased fish did not reveal the presence of any parasites.

### 3.2. Pathological Features

Histopathological analysis of the diseased fish revealed severe pathological changes in the liver, kidney, and spleen, including extensive necrosis, vacuolation, and focal hemorrhaging (Figure 2). The liver lesions were marked by hepatocyte necrosis and significant vacuolation (white arrow; Figure 2B). Some cell boundaries were not clear, and the membrane was disintegrated or even absent. Additionally, there was pronounced swelling and hemorrhaging in the liver blood sinuses (red arrow and blue triangular arrowheads; Figure 2B). The infected kidney tissues showed edema in the renal tubules (white arrow; Figure 2D) accompanied by lymphocyte infiltration in and around the affected areas (blue triangular arrowheads; Figure 2D). The spleen exhibited varying degrees of necrosis, vacuolation (white arrow; Figure 2F), and enlargement of melanomacrophage centers (blue triangular arrowheads; Figure 2F).

### 3.3. Morphological Characterization of WH10

The colonies of isolate WH10 were cultured on BHI agar, displaying a yellow hue. The surface of the colony appeared to be moist and glossy. The bacterium was Gram-negative, and was characterized by a rod-shaped structure, with no budding cells or pods (Figure 3A). Under scanning electron microscopy, the bacterium appeared arc-shaped and was approximately 1 µm in length (Figure 3B). 

### 3.4. Sequence Analysis of the 16S rRNA Gene

BLAST searching in the NCBI GenBank database indicated that the identified sequences had significant similarity to *Aeromonas veronii* 16S rRNA gene sequences. The constructed phylogenetic tree showed that the WH10 isolate was closely related to *Aeromonas veronii*, thus confirming its taxonomic classification as *Aeromonas veronii* at the molecular level (Figure 4A).

### 3.5. Virulence Gene Detection

Herein, we screened the PCR profiles of ten virulence genes, which indicated the presence of the alt, act, lip, ascV, and LuxS genes in isolate WH10, while the remaining five genes were not detected (Figure 4B). 

### 3.6. Bacterial Biochemical Identification

WH10 was subjected to biochemical tests using a biochemical analyzer (Table 2). WH10 was positive for N-Acetyl-D-Glucosamine, Dextrin, Sucrose, 1% NaCl, Acidic PH PH6, Niaproof 4, D-Galactonic Acid, D-Glucose-6-po4, Vancomycin, Nalidixic Acid, Rifamycin SV, Tween 40 reaction, Tetrazolium Blue, L-Malic Acid, and D-Fructose-6-po4, but was negative for Acidic PH PH5, D-Malic Acid, P-Hydroxy-Phenylacetic Acid, Tetrazolium Violet, N-Acetyl-D-Galactosamine, 4% NaCl, 8% NaCl, D-Ducrose, Inosine, Fusidic Acid, 3-Methyl Glucose, Lincomycin, D-Glucuronic Acid, Glucuronamide, D-Arabitol, D-Aspartic acid, Troleandomycin, Minocycline, Potassium Tellurite, Sodium Bromate, Sodium Butyrate, Formic Acid, Propionic Acid, and α-Keto-Butyric Acid (Table 2). According to the morphological and biochemical results, isolate WH10 was identified tentatively as *Aeromonas veronii*. Owing to the intricate diversity within *Aeromonas veronii* members and the restricted number of biochemical reactions assessed by the Biolog analysis system, the test successfully identified this isolate as the species *Aeromonas veronii*.

### 3.7. Antibiotic Susceptibility Analysis

In the antibiotic susceptibility analysis, WH10 showed varying susceptibilities to ten antibacterial drugs. Notably, it was sensitive to four antibacterial drugs: compound sulfamethoxazoles, cefothiophene, doxycycline, and sulfamethoxazole (Table 3).

### 3.8. Differential Leukocyte Counts (DLC)

As depicted in Figure 5, infection with WH10 significantly impacted the blood parameters of bronze gudgeon. In the affected group, the percentage of neutrophils (50.3%) was significantly higher than that of the control group (*p* < 0.01). The percentage of monocytes also increased in infected group, reaching 20.7% (*p* < 0.01), in comparison with that of the control group. Conversely, the lymphocyte percentage in the infected group (27.3%) was markedly lower than that in the controls (*p* < 0.01). Furthermore, throughout the experimental process, the percentage of eosinophils in the infected group remained at basal levels compared with those in the control group (Figure 5A).

### 3.9. Serum Biochemical Analysis

The comparison of the serum biochemical parameters between diseased and healthy bronze gudgeon is shown in Figure 5. Diseased bronze gudgeon had significantly elevated AST (148 g/L) activity and TG (4.1 g/L) concentration compared with those of the healthy controls (*p* < 0.01) (Figure 5B,D). Conversely, the concentrations of TP (24.2 g/L), ALB (5.0 g/L), T-CHO (3.8 g/L), and GLU (1.5 g/L) were notably reduced in the infected bronze gudgeon compared with those in the healthy specimens (Figure 5C,D). There was no significant difference in the ALT (7.3 g/L) and ALP (22.7 g/L) activities between the two groups (Figure 5B).

### 3.10. Pathogenicity

The challenge of healthy bronze gudgeon with purified isolate WH10 resulted in hemorrhagic signs reminiscent of those observed in naturally infected fish at around 4 days post-infection (dpi). Within 12 dpi, mortality approached 80%. The pathogenicity test demonstrated mortality in all groups of fish exposed to varying concentrations of bacteria. However, 80% survival was recorded among individuals in the experimental group receiving 2 × 10^8^ CFU/mL fish weight. Conversely, no sign was recorded in the mock-infected fish. According to the experimental results, the LD_50_ of WH10 was determined as 1.36 × 10^6^ CFU/mL (Figure 6). Samples from challenged and dead fish tested positive for WH10 by PCR. These results unequivocally confirmed that WH10, isolated from infected bronze gudgeon, was the etiological agent of the disease observed in bronze gudgeon farms.

## 4. Discussion

*Aeromonas veronii*, a motile, rod-shaped, and Gram-negative bacterium, belongs to the *Aeromonadaceae* family. It exhibits a broad host range and has the capacity to induce zoonotic motile *Aeromonas* septicemia (MAS) in humans, terrestrial animals, and various aquatic species [18,19]. *Aeromonas veronii* infections are particularly lethal to aquatic animals, frequently leading to significant outbreaks and high mortality rates in aquaculture [20]. Reports indicate that *Aeromonas* outbreaks resulted in a 10% loss in fish farm production in 2017 [21]. Among pathogenic *Aeromonas* species, such as *Aeromonas hydrophila*, *Aeromonas caviae*, *Aeromonas sobria*, and *Aeromonas veronii*, the incidence of *Aeromonas veronii* infections has notably increased in recent years, leading to frequent isolation of these pathogens from diseased fish. In recent years, *Aeromonas veronii* has been documented in Nile tilapia (*Oreochromis niloticus*) [22], Chinese Soft-Shelled Turtle (*Trionyx sinensis*) [12], channel catfish (*Ictalurus punctatus*) [23], crucian carp (*Carassius auratus*) [24], Rohu (*Labeo rohita*) [25], and rainbow trout (*Oncorhynchus mykiss*) [26]. The clinical manifestations of infected fish often include systemic hemorrhagic septicemia and cutaneous ulcers. In our study, the clinical signs observed in diseased bronze gudgeon comprised hemorrhages on the body surface and fins, along with ulcerated skin lesions. Visceral tissue, including the liver and spleen, exhibited hemorrhagic lesions, which were similar to those commonly observed in fish species infected with *Aeromonas veronii*. In addition, the pathogenicity test showed that the LD_50_ was 1.36 × 10^6^ CFU/mL per fish weight, indicating that WH10 exhibited strong virulence towards bronze gudgeon. Furthermore, *Aeromonas veronii* was successfully detected and isolated from bronze gudgeon artificially infected with WH10. Together, our results indicated that *Aeromonas veronii* WH10 was the pathogenic bacterium responsible for the observed disease in bronze gudgeon. 

Hematology has been used frequently to monitor the health status and immunological responses of aquaculture species [27,28]. Monocytes and neutrophils, serving as the main phagocytic cells in fish, are integral components of the non-specific defense system [29]. In our study, the serum neutrophil and monocyte percentage of *Aeromonas veronii*-infected bronze gudgeon were increased significantly. This finding might indicate enhanced phagocytosis in the infected bronze gudgeon, which is related to biological responsiveness to environmental parameters and individual health status [30]. Similar alterations in blood cell proportions post-bacterial infection were observed in fancy carp (*Cyprinus carpio*) infected with *Aeromonas veronii* [31] and in *Vibrio metschnikovii*-infected hybrid sturgeon [32]. However, in *Aeromonas hydrophila*-infected tilapia, numbers of circulating neutrophils decreased, together with augmented accumulation of granulocytes at the inflammatory site [33]. These results differed from those reported herein, which might be related to the fish species and the pathogen under evaluation. These contrasting results might also reflect the fact that a high number of neutrophils in circulation is caused by their delayed apoptosis or impaired clearance from the infection site, thereby contributing to the prolonged secretion of pro-inflammatory mediators, e.g., bacterial components, interleukin-8, or granulocyte macrophage colony-stimulating factor [33]. Lymphocytes are crucial for the body’s cellular immune response, and lower lymphocyte levels are indicative of compromised immune function. Herein, serum lymphocyte levels were decreased significantly in infected bronze gudgeon, indicating a weakened immune system. 

Additionally, in fish, the liver is an important organ for protein synthesis, and impairment of hepatic cells leads to a decline in protein production [34]. Serum ALB and TP serve as markers of the immune response and hepatocyte injury, respectively. The observed reduction in serum ALB and TP concentrations in diseased bronze gudgeon signified hepatocyte damage and compromised liver function. Notably, AST is predominantly localized in the cytoplasm of hepatocytes, and its release into the bloodstream occurs exclusively during liver inflammation, necrosis, or intoxication [35]. In this study, diseased bronze gudgeon exhibited a notable increase in serum AST activity (*p* < 0.01), indicating potential hepatic damage in these individuals. Furthermore, a decrease in T-CHO was observed in the serum of infected bronze gudgeon, suggestive of malnutrition, likely caused by food refusal and starvation during the experimental phase. Pathological alterations in fish kidney, including changes in renal tubules and impaired glomerular filtration function, negatively impact the physiological metabolism of the affected fish [36]. Our findings revealed decreased GLU concentration in diseased fish relative to the control group, suggesting disturbances in glomerular filtration and reabsorption functions after infection with WH10.

Drug sensitivity testing serves as a crucial step to identify effective disease treatment options [37]. Testing the antibiotic response of WH10 demonstrated its sensitivity to compound sulfamethoxazoles, cefothiophene, doxycycline, and sulfamethoxazole. It exhibited moderate sensitivity to gentamicin, amikacin, neomycin, florfenicol, and enrofloxacin, while displaying resistance to ciprofloxacin. The test results were different from those reported previously for *Aeromonas veronii* [12,38,39]. This variability in antibiotic sensitivity might be attributed to differences in the strains’ origins or their distinct antibiotic resistance profiles. The resistance of *Aeromonas veronii* to multiple antibiotics represents a significant health concern for both animals and humans. Improper use of antibiotics in aquaculture and the potential for *Aeromonas veronii* to acquire resistance genes from other multidrug-resistant pathogens are major factors contributing to the emergence of drug-resistant strains [12]. Therefore, selecting antibiotics based on the results of susceptibility testing is crucial for the effective treatment of bacterial diseases.

The pathogenicity of *Aeromonas* spp. is enhanced by their ability to produce various virulence factors, including proteases, hemolysin, enterotoxins, and toxins in aquatic environments. These virulence factors contribute to the bacteria’s ability to adhere to, invade, and destroy host cells, thereby overcoming the host’s immune response [40]. Assessing the pathogenicity and virulence of a bacterial pathogen according to the presence of virulence factors is essential [41]. In this study, we investigated virulence-related factors associated with *Aeromonas* species infection, including alt, aha, act, lip, ast, ascV, exu, LuxS, aerA, and hlyA. The ast, alt, and act genes encode cytotoxic enterotoxins, representing critical virulence determinants for *Aeromonads*, which are potent foodborne pathogens. Knockout mutants of the act gene showed significantly reduced virulence [42]. Herein, *Aeromonas veronii* WH10 contained act and alt genes, and exhibited a beta hemolysis reaction on blood agar. Additionally, virulence genes encoding secreted proteins and toxins, such as the lip gene, and the gene encoding the quorum sensing signal molecule AI-2-LuxS, are major pathogenic factors of *Aeromonas veronii* [43,44], contributing to bacterial biofilm production and virulence [45,46]. *Aeromonas veronii* WH10 contained lip and LuxS genes, suggesting that its high pathogenicity was caused by a combination of multiple virulence-related factor. Our results suggested that an *Aeromonas veronii* detection method, and even a multivalent vaccine, could be effectively constructed based on its mutual virulence genes.

## 5. Conclusions

This study identified *Aeromonas veronii* strain WH10 as the agent that caused an emerging disease in farmed bronze gudgeon in China. This strain was tentatively named WH10 based on its morphology, phylogeny, clinical signs, pathology, and the results of the pathogen challenge. The isolation and identification of WH10 from bronze gudgeon expands our understanding of the host spectrum of *Aeromonas veronii*. Additionally, this study lays the foundation to develop diagnostic methods for bronze gudgeon disease and to investigate the invasion, transmission mechanisms, and potential therapeutic interventions for related pathogens.

## Figures and Tables

**Figure 1 animals-14-02440-f001:**
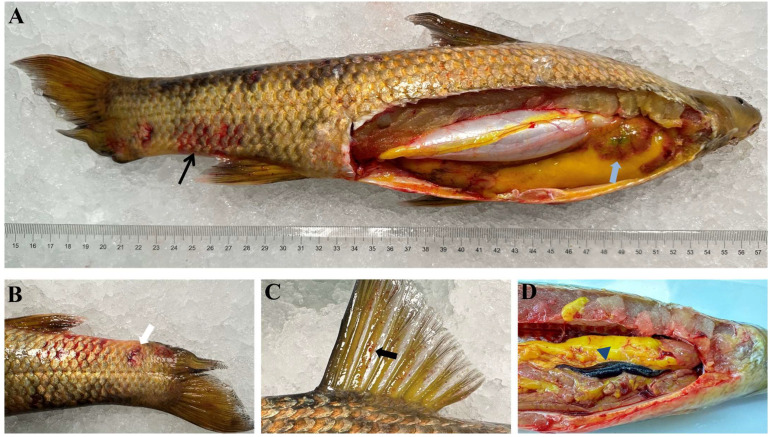
Clinical manifestations of infected bronze gudgeon. (**A**) The infected bronze gudgeon showed hemorrhages on both body surface (black arrow) and liver (blue arrow). (**B**) Ulcerated skin was evident on the surface of the affected bronze gudgeon (white arrow). (**C**) Bleeding points were observed on the fin plates (black arrow). (**D**) The spleen of the diseased bronze gudgeon appeared dark and enlarged (navy blue arrowheads).

**Figure 2 animals-14-02440-f002:**
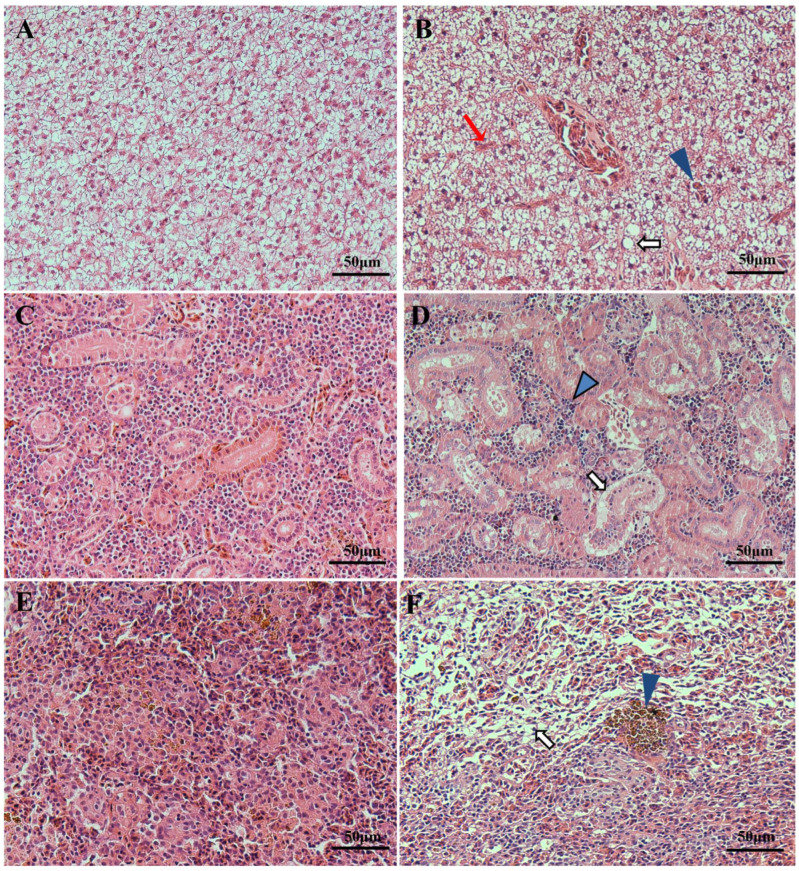
Pathological analysis of the diseased bronze gudgeon. (**A**) Healthy liver. (**B**) Significant vacuolation (white arrow), along with liver blood sinuses exhibiting swelling and hemorrhaging, was observed in the diseased liver tissue (red arrow and blue triangular arrowheads). (**C**) Healthy kidney. (**D**) Infected kidney tissues exhibited edema in the renal tubules (white arrow) and lymphocyte infiltration in and around the affected areas (blue triangular arrowheads). (**E**) Healthy spleen. (**F**) The spleen displayed a spectrum of extensive necrosis and vacuolation (white arrow) and extended melanomacrophage center (blue triangular arrowhead).

**Figure 3 animals-14-02440-f003:**
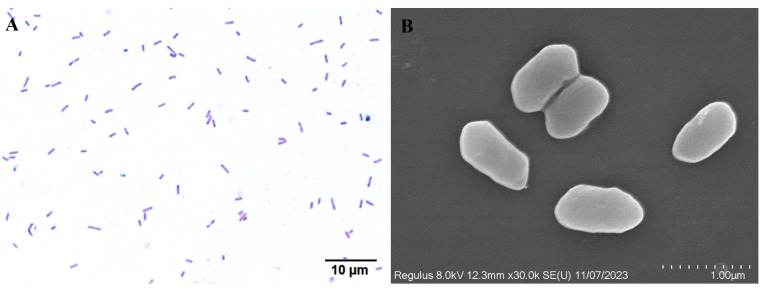
Scanning electron microscopy and Gram staining of *Aeromonas veronii* (WH10). (**A**) Gram staining of isolate WH10 (scale bar: 10 µm); (**B**) Scanning electron micrograph of isolate WH10 (scale bar: 1.0 µm).

**Figure 4 animals-14-02440-f004:**
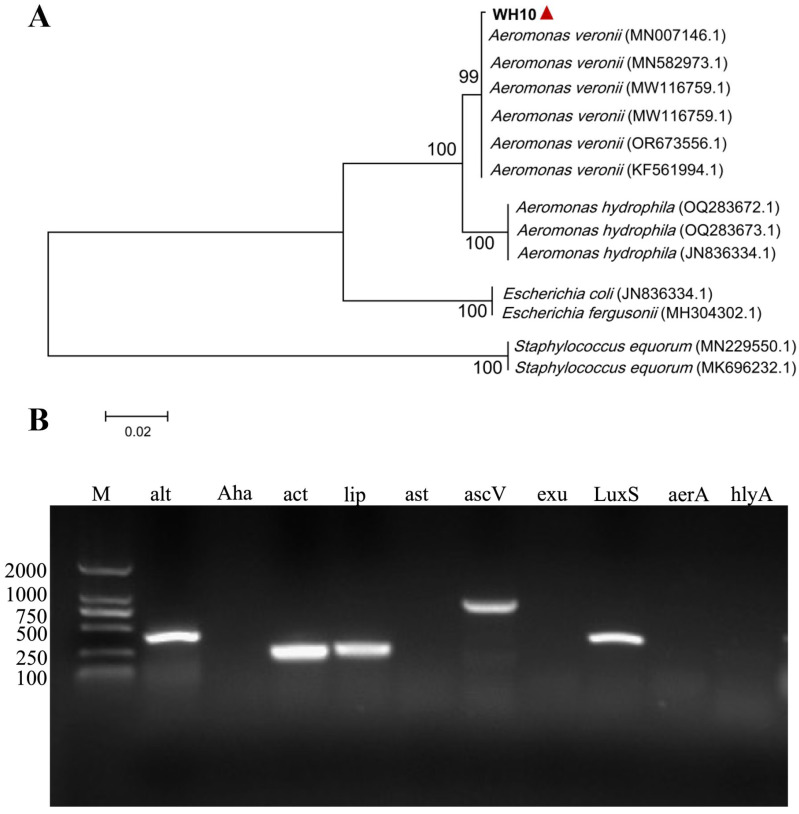
Phylogenetic analysis and PCR detection of virulence genes of isolate WH10. (**A**) 16S rRNA-based phylogenetic analysis using sequences from *Aeromonas veronii*, *Aeromonas hydrophila*, *Escherichia coli*, *Escherichia fergusonii*, and *Staphylococcus equorum*. Red arrow is the strain isolated in our study. (**B**) Agarose gel electrophoresis of virulence genes from the WH10 strain. M: DNA ladder (DL2000 bp); 1st lane: alt; 2nd lane: aha; 3rd lane: act; 4th lane: lip; 5th lane: ast; 6th lane: ascV; 7th lane: exu; 8th lane: LuxS; 9th lane: aerA; 10th lane: hlyA.

**Figure 5 animals-14-02440-f005:**
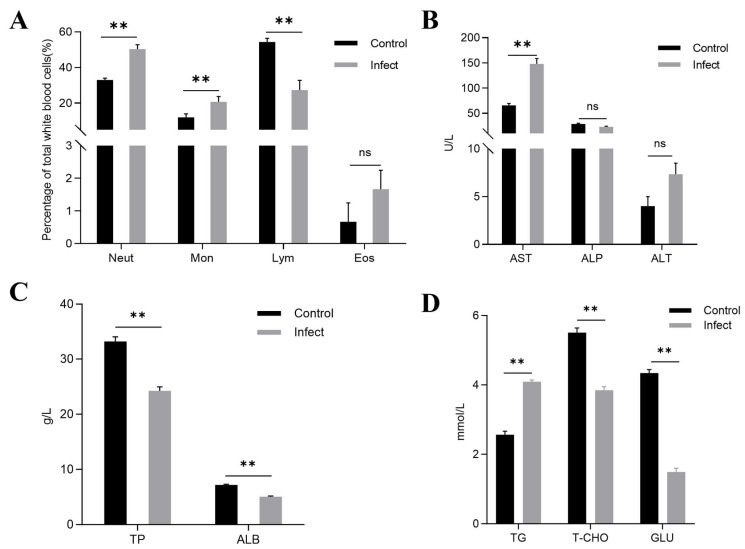
Differential leukocyte counts and biochemical evaluations of serum from bronze gudgeons infected with isolate WH10. (**A**) The percentage changes in differential leukocyte counts in the affected group compared to the control group. (**B**) Serum biochemical evaluations of affected bronze gudgeon in showed significantly higher AST, but no significant difference in ALP activity and ALT concentration compared with those in the serum of healthy fish. (**C**) Significantly lower TP and ALB concentrations in affected fish serum compared with those in the serum of healthy fish. (**D**) T-CHO and GLU concentrations were significantly lower in the serum of affected fish lower than in that of healthy fish. ** *p* < 0.01. ns: not significant.

**Figure 6 animals-14-02440-f006:**
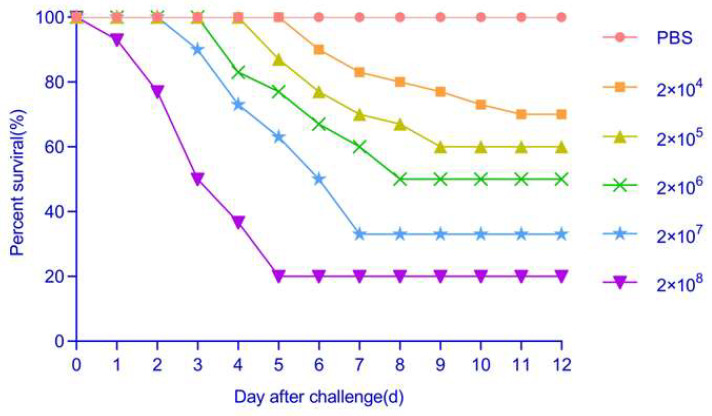
Pathogenicity of different concentrations of the WH10 strain solution toward bronze gudgeon.

**Table 1 animals-14-02440-t001:** Primer sequences used for PCR amplification of target genes of *Aeromonas veronii*.

Gene Name	Primer Name	Primers (5′-3″)	References
16S rRNA	16S rRNA-F	AGAGTTTGATCATGGCTCAG	[12]
	16S rRNA-R	TACGGTTACCTTGTTACGACTT	
ascV	ascV-F	AGCAGATGAGTATCGACGG	[13]
	ascV-R	AGGCATTCTCCTGTACCAG	
hlyA	hlyA-F	GGCCGGTGGCCCGAAGATACGGG	[14]
	hlyA-R	GGCGGCGCCGGACGAGACGGGG	
aerA	aerA-F	CAAGAACAAGTTCAAGTGGCCA	[13]
	aerA-R	ACGAAGGTGTGGTTCCAGT	
LuxS	LuxS-F	GATCCTCTCCGAGGCGTGG	[14]
	LuxS-R	AGGCTTTTCAGCTTCTCTTCC	
act	act-F	TCTCCATGCTTCCCTTCCACT	[13]
	act-R	AACTGACATCGGCCTTGAACTC	
alt	alt-F	TGACCCAGTCCTGGCACGGC	[13]
	alt-R	GGTGATCGATCACCACCAGC	
ast	ast-F	TCTCCATGCTTCCCTTCCACT	[13]
	ast-R	GTGTAGGGATTGAAGAAGCCG	
lip	lip-F	CACCTGGT(T/G)CCGCTCAAG	[15]
	lip-R	GTACCGAACCAGTCGGAGAA	
exu	exu-F	AGACATGCACAACCTCTTCC	[16]
	exu-R	GATTGGTATTGCCTTGCAAG	
aha	aha-F	GGCTATTGCTATCCCGGCTCTGTT	[15]
	aha-R	CGGTCCACTCGTCGTCCATCTTG	

**Table 2 animals-14-02440-t002:** Biochemical identification of the WH10 strain.

	Reaction Item	Result *		Reaction Item	Result *
A1	Negative control	N	E1	Gelatin	B
A2	Dextrin	P	E2	Glycyl-L-Proline	B
A3	D-Maltose	B	E3	L-Alanine	B
A4	D-Trehalose	B	E4	L-Arginine	B
A5	D-Cellobiose	B	E5	L-Aspartic Acid	B
A6	Gentiobiose	B	E6	L-Glutamic Acid	B
A7	Sucrose	P	E7	L-Histidine	B
A8	D-Turanose	B	E8	L-Pyroglutamic Acid	B
A9	Stachyose	B	E9	L-Serine	B
A10	Positive control	P	E10	Lincomycin	N
A11	Acidic PH PH6	P	E11	Guanidine HCl	B
A12	Acidic PH PH5	N	E12	Niaproof 4	P
B1	D-Raffinose	B	F1	Pectin	B
B2	α-D-Lactose	B	F2	D-Galacturonic Acid	B
B3	D-Melibiose	B	F3	L-Galactonic Acid Lactone	B
B4	β-Methyl-D-Glucoside	B	F4	D-Galactonic Acid	P
B5	D-Salicin	B	F5	D-Glucuronic Acid	N
B6	N-Acetyl-D-Glucosamine	P	F6	Glucuronamide	N
B7	N-Acetyl-β-Mannosamine	B	F7	Mucic Acid	B
B8	N-Acetyl-D-Galactosamine	N	F8	Quinic Acid	B
B9	N-Acetyl Neuraminic	B	F9	D-Saccharic Acid	B
B10	1% NaCl	P	F10	Vancomycin	P
B11	4% NaCl	N	F11	Tetrazolium Violet	N
B12	8% NaCl	N	F12	Tetrazolium Blue	P
C1	α-D-Glucose	B	G1	P-Hydroxy-Phenylacetic Acid	N
C2	D-Mannose	B	G2	Methyl Pyruvate	B
C3	D-Fructose	B	G3	D-Lactic Acid Methyl Ester	B
C4	D-Galactose	B	G4	Lactic Acid	B
C5	3-Methyl Glucose	N	G5	Citric Acid	B
C6	D-Ducrose	N	G6	α-Keto-Glutaric Acid	B
C7	L-Fucose	B	G7	D-Malic Acid	N
C8	L-Rhamnose	B	G8	L-Malic Acid	P
C9	Inosine	N	G9	Bromo-Succinic-Acid	B
C10	1% Sodium Lactate	B	G10	Nalidixic Acid	P
C11	Fusidic Acid	N	G11	Lithium Chloride	B
C12	D-Serine	B	G12	Potassium Tellurite	N
D1	D-Sorbitol	B	H1	Tween 40	P
D2	D-Mannitol	B	H2	γ-Amino-Butyric-Acid	B
D3	D-Arabitol	N	H3	α-Hydroxy- Butyric-Acid	B
D4	Myo-Inositol	B	H4	β--Hydroxy-D, L Butyric-Acid	B
D5	Glycerol	B	H5	α-Keto-Butyric Acid	N
D6	D-Glucose-6-po4	P	H6	Acetoacetic Acid	B
D7	D-Fructose-6-po4	P	H7	Propionic Acid	N
D8	D-Aspartic acid	N	H8	Acetic Acid	B
D9	D-Serine	B	H9	Formic Acid	N
D10	Troleandomycin	N	H10	Aztreonam	B
D11	Rifamycin SV	P	H11	Sodium Butyrate	N
D12	Minocycline	N	H12	Sodium Bromate	N

* Abbreviations: P, positive; N, negative; B, borderline.

**Table 3 animals-14-02440-t003:** Antibiotic susceptibility analysis of the WH10 strain.

Drugs	Content (μg/Piece)	Inhibition Zone (mm)	Sensitivity
Compound Sulfamethoxazoles	23.75	29	S
Cefothiophene	30	24	S
Doxycycline	30	19	S
Sulfamethoxazole	300	16	S
Gentamicin	10	13	I
Amikacin	30	14	I
Neomycin	30	14	I
Florfenicol	30	15	I
Enrofloxacin	10	13	I
Ciprofloxacin	5	8	R

Notes: “S” = Susceptible (diameter > 15 mm); “I” = Intermediate (10 mm < diameter ≤ 15 mm); “R” = Resistant (diameter ≤ 10 mm).

## Data Availability

The original contributions presented in the study are included in the article, further inquiries can be directed to the corresponding author/s.

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
