# Peer review of "Identification of Aeromonas veronii as the Pathogen Associated with Massive Mortality in Bronze Gudgeon (Coreius heterodon)"

_animals, 2024, doi:10.3390/ani14162440_

Round 1

Reviewer 1 Report

Comments and Suggestions for Authors

Very poorly written manuscript. Needs thorough revision before it can be considered for publication.

Introduction: The author should describe the importance of disease, particularly bacterial diseases, in aquaculture and how A. veronii is contributing to significant mortality in aquaculture systems.

Materials and Method: What are the methods used to ensure the fish species used in the experiment were free from pathogens? 

Apart from bacterial analysis, analysis of viral, fungal, and parasitic pathogens was done.

Whether Koch's postulates were followed by reisolation of bacterial pathogens from challenged fish.

Results: Histology, the liver figures in control, and treatment look same (not much difference).

It is better to provide figures for differential leukocyte counts.

The discussion and conclusion need modifications.

References:

Cite papers related to A. veronii infection in fish. e.g., Behera, B.K., Parida, S.N., Kumar, V., Swain, H.S., Parida, P.K., Bisai, K., Dhar, S. and Das, B.K., 2023. Aeromonas veronii is a lethal pathogen isolated from gut of infected Labeo rohita: molecular insight to understand the bacterial virulence and its induced host immunity. Pathogens12(4), p.598.

Comments on the Quality of English Language

Needs major revision

Author Response

Reviewer #1:   

  1. Very poorly written manuscript. Needs thorough revision before it can be considered for publication.

Response: Thank you for your comments. We have carefully reviewed the entire manuscript and identified areas where the language needs refinement for more precise expression. Additionally, we engaged native English-speaking scientists at Elixigen Company (Huntington Beach, California) to revise our manuscript. The relevant language polishing certificate has been submitted in the review system. We believe these revisions will enhance the clarity and conciseness of the text, making it more comprehensible and impactful for future publication.

  1. Introduction: The author should describe the importance of disease, particularly bacterial diseases, in aquaculture and how A. veronii is contributing to significant mortality in aquaculture systems.

Response: Thank you for your valuable suggestions and comments. Because we would like to further analyze the harmfulness of Aeromonas veronii in aquaculture and compare it with the WH10 strain isolated in this study, we have included the detailed description of Aeromonas veronii in the Discussion section instead of the Introduction. This placement allows for a more in-depth comparison and contextual analysis within the framework of our findings. While we initially discussed the importance of Aeromonas veronii in the Discussion section, we acknowledge that the coverage was insufficient. Therefore, we have expanded the discussion on Aeromonas veronii in the manuscript to provide a more comprehensive analysis and address the reviewers' concerns.

Below is the revised and more detailed section on the characteristics of Aeromonas veronii, its impact on aquaculture fish, and associated mortality rates. “Aeromonas veronii, a motile, rod-shaped, and gram-negative bacterium, belongs to the Aeromonadaceae family (Hickman-Brenner et al., 1987). It exhibits a broad host range and has the capacity to induce zoonotic motile Aeromonas septicemia (MAS) in humans, terrestrial animals, and various aquatic species (Rahman et al., 2002; Cui et al., 2017). Aeromonas veronii infections are particularly lethal to aquatic animals, frequently leading to significant outbreaks and high mortality rates in aquaculture (Fernández-Bravo et al., 2020). Reports indicate that Aeromonas outbreaks resulted in a 10% loss in fish farm production in 2017 (Abd-El-Malek, 2017). Among pathogenic Aeromonas species such as Aeromonas hydrophila, Aeromonas caviae, Aeromonas sobria, and Aeromonas veronii, the incidence of Aeromonas veronii infections has notably increased in recent years, leading to frequent isolation of these pathogens from diseased fish. In recent years, Aeromonas veronii has been documented in Nile tilapia (Oreochromis niloticus) (Dos Santos et al., 2023), Chinese Soft-Shelled Turtle (Trionyx sinensis) (Hu et al., 2023), channel catfish (Ictalurus punctatus) (Haitham et al., 2018), freshwater goldfish (Carassius auratus) (Chen et al., 2019), Labeo rohita (Behera et al., 2023), and rainbow trout (Oncorhynchus mykiss) (Reyes-Rodríguez et al., 2019). The pathogenicity of Aeromonas spp. is enhanced by their ability to produce various virulence factors, including proteases, hemolysin, enterotoxins, and toxin genes of Aeromonas spp. in aquatic environments. These virulence factors contribute to the bacteria's ability to adhere to, invade, and destroy host cells, thereby overcoming the host's immune response (Majeed et al., 2023).” (Lines 331-346).

  1. Materials and Method: What are the methods used to ensure the fish species used in the experiment were free from pathogens? Apart from bacterial analysis, analysis of viral, fungal, and parasitic pathogens was done.

Response: Thank you very much for concerning. The healthy bronze gudgeon was sourced from the breeding station of Institute of Hydroecology, MWR & CAS, which has not been reported any instances of bronze gudgeon infected with Aeromonas veronii. The fish were transported to the laboratory in oxygenated bags to ensure their health during transport to the laboratory for subsequent experiments. Prior to the experiment, the bronze gudgeon was observed for seven days to monitor their health condition. Meanwhile, a comprehensive pathogen screening, including tests for bacteria, viruses, fungi, and parasites, was conducted, with all results showing negative, thus confirming the pathogen-free status of the fish. In addition, during the experiment, any diseased bronze gudgeon was promptly sampled and treated to prevent the spread of pathogens. These measures collectively ensure, to a significant extent, that the bronze gudgeon used in the experiment were free from pathogens.

  1. Whether Koch's postulates were followed by re-isolation of bacterial pathogens from challenged fish.

Response: Many thanks. Regarding adherence to Koch's postulates for re-isolating bacterial pathogens, we have incorporated this step into Results of our manuscript. After artificially infecting healthy bronze gudgeon, bacteria were re-isolated from the symptomatic fish. These bacteria were identified using molecular biological methods and confirmed to be the Aeromonas veronii WH10 strain, matching the initial infection strain. This confirmed the pathogenicity of the strain and fulfilled Koch's postulates for pathogen re-isolation in our experiments.

  1. Results: Histology, the liver figures in control, and treatment look same (not much difference).

Response: Thank you very much for concerning. We have re-examined the liver tissue sections and marked the differences between the control and infected groups with triangles or arrows to clearly highlight these differences (Figure 2B). Furthermore, we have carefully observed and analyzed the pathological sections of other tissues to provide more comprehensive pathological information.

  1. It is better to provide figures for differential leukocyte counts.

Response: Thank you very much for your friendly comments. Our manuscript has presented the relevant data and graphical results of the differential white blood cell counts (Figure 5A). In results of our manuscript, Lines 300-307 detailed the changes in blood parameters of bronze gudgeon infected with WH10, including the percentage variations in neutrophils, monocytes, lymphocytes, and eosinophils. These changes were also visually depicted in Figure 5A, clearly illustrating the differences between the infected and control groups.

  1. The discussion and conclusion need modifications.

Response: Thanks. Based on your comments, we have carefully revised and enhanced the discussion and conclusion sections. In the discussion, we further analyzed the significance of our experimental results, incorporating relevant research literature to strengthen comparisons and discussions with existing studies. We have highlighted the changes in the review manuscript (manuscript with highlighted copy). In the conclusion, we have succinctly summarized the main findings and emphasized the importance of our study for the prevention and control of diseases in bronze gudgeon.

  1. References:

Cite papers related to A. veronii infection in fish. e.g., Behera, B.K., Parida, S.N., Kumar, V., Swain, H.S., Parida, P.K., Bisai, K., Dhar, S. and Das, B.K., 2023. Aeromonas veronii is a lethal pathogen isolated from gut of infected Labeo rohita: molecular insight to understand the bacterial virulence and its induced host immunity. Pathogens, 12(4), p.598.

Response: We sincerely thank the reviewer for thoroughly examining our manuscript and providing insightful comments that guided our revisions. We carefully reviewed the article, accurately cited sources, and incorporated the suggested references into the discussion section (Lines 345). Additionally, we compared data for compatibility as per your proposal, enhancing the robustness of our findings.

Behera, B. K., Parida, S. N., Kumar, V., Swain, H. S., Parida, P. K., Bisai, K., Dhar, S., & Das, B. K. (2023). Aeromonas veronii Is a Lethal Pathogen Isolated from Gut of Infected Labeo rohita: Molecular Insight to Understand the Bacterial Virulence and Its Induced Host Immunity. Pathogens, 12, 598.

Reviewer 2 Report

Comments and Suggestions for Authors

The paper by Liu, Li, Fan, and colleagues is interesting and well written, and presents an intriguing link between their discovered Aeromonas veronii pathogen, WH10, and the mortality of an economically important fish species in China. The findings are intriguing and the disease-association, in particular, makes for a compelling story. While I think that the message of the paper is certainly of interest for the readers of Animals, there are some minor issues in the manuscript that need to be improved.

 1. Aeromonas veronii is a very common and destructive bacteria in modern aquaculture, causing very large aquaculture losses. What do you think effective strategies to mitigate its impact and reduce harm in bronze gudgeon aquaculture?2. In this study, the authors chose ten virulence-related genes—Aha, exu, lip, ast, alt, act, LuxS, aerA, hlyA, and ascV—for amplifying Aeromonas veronii using conventional PCR with template DNA from the WH10 strain. What was the rationale behind selecting these specific genes to assess WH10's virulence? Is there empirical evidence supporting their relevance in determining bacterial virulence? 3. Furthermore, it is important to ensure uniformity in formatting the names of virulence genes. Specifically, some genes are italicized while others are written in regular font. Please ensure consistent and correct formatting of all gene names according to scientific conventions.

4. For the experiments of bacterial isolation and identification, it is recommended to provide more background information about sample collection and processing, such as the number of samples collected, sources and storage conditions, so as to enhance the repeatability of the experiments.

5. In the analysis of blood parameters, it is essential for the authors to give a detailed description of the time and method of blood sample collection. Additionally, they should outline measures taken to ensure the stability and accuracy of the samples throughout the process.

6. In conjunction with the antibiotic sensitivity results of Aeromonas veronii in this study, the discussion should explore potential causes and influencing factors by comparing and analyzing findings from related studies.

7. Figure 1: It is recommended to adjust the color of the indicating arrow on clinically diseased fish to improve visibility against the background. For instance, in Figure 1B and D, the arrow color blends with the background, making it challenging to discern clearly.

Author Response

Reviewer #2:

The paper by Liu, Li, Fan, and colleagues is interesting and well written, and presents an intriguing link between their discovered Aeromonas veronii pathogen, WH10, and the mortality of an economically important fish species in China. The findings are intriguing and the disease-association, in particular, makes for a compelling story. While I think that the message of the paper is certainly of interest for the readers of Animals, there are some minor issues in the manuscript that need to be improved.

  1. Aeromonas veronii is a very common and destructive bacteria in modern aquaculture, causing very large aquaculture losses. What do you think effective strategies to mitigate its impact and reduce harm in bronze gudgeon aquaculture?

Response: Many thanks for your valuable comment. Several measures are commonly adopted to prevent and control Aeromonas veronii infections. Firstly, the development of specific vaccines against Aeromonas veronii is an effective preventive approach. Secondly, Chinese herbs medicine, known for their antibacterial, anti-inflammatory, and immunomodulatory effects, play a significant role. Chinese herbs medicines such as Scutellaria, Coptis, and Phellodendri contain various active ingredients that inhibit the growth and reproduction of Aeromonas veronii. Additionally, some Chinese herbs can enhance the immunity of bronze gudgeon, regulate immune balance, and improve resistance to pathogens.

In practical applications, combining vaccination with Chinese herbs can be particularly effective. Vaccination establishes basic immune protection for fish at the early breeding stage. During the breeding process, adding appropriate amounts of Chinese herbs to the feed regularly helps maintain the health and immune function of the fish. Moreover, strengthening the management of the breeding environment, optimizing water quality, and maintaining appropriate breeding density are crucial steps in reducing the harm caused by Aeromonas veronii in aquaculture.

Overall, the development of vaccines and the use of Chinese herbal medicine hold great promise for the prevention and control of Aeromonas veronii. These strategies are expected to significantly reduce the impact of Aeromonas veronii on bronze gudgeon farming. Here are several articles on the effectiveness of vaccines and Chinese herbal medicines in controlling Aeromonas veronii in aquaculture:

[1] Jiao, X., Zhang, D. X., Chen, C., Kong, L. C., Hu, X. Y., Shan, X. F., & Qian, A. D. (2023). Immunization effect of recombinant Lactobacillus casei displaying Aeromonas veronii Aha1 with an LTB adjuvant in carp. Fish & shellfish immunology, 135, 108660.

[2] Chi, Y., Jiao, H., Ran, J., Xiong, C., Wei, J., Ozdemir, E., & Wu, R. (2023). Construction and efficacy of Aeromonas veronii mutant Δhcp as a live attenuated vaccine for the largemouth bass (Micropterus salmoides). Fish & shellfish immunology, 136, 108694.

[3] Youssef, H. A., Ayoub, H. F., Soror, E. I., & Matter, A. F. (2023). Virulence genes contributing to Aeromonas veronii pathogenicity in Nile tilapia (Oreochromis niloticus): approaching the development of live and inactivated vaccines. Aquaculture international: journal of the European Aquaculture Society, 31(3), 1253–1267.

[4] Song, H., Zhang, S., Yang, B., Liu, Y., Kang, Y., Li, Y., Qian, A., Yuan, Z., Cong, B., & Shan, X. (2022). Effects of four different adjuvants separately combined with Aeromonas veronii inactivated vaccine on haematoimmunological state, enzymatic activity, inflammatory response and disease resistance in crucian carp. Fish & shellfish immunology, 120, 658–673.

[5] Song, H. C., Yang, Y. X., Lan, Q. G., & Cong, W. (2023). Immunological effects of recombinant Lactobacillus casei expressing pilin MshB fused with cholera toxin B subunit adjuvant as an oral vaccine against Aeromonas veronii infection in crucian carp. Fish & shellfish immunology, 139, 108934.

[6] Li, T., Wang, Z., Han, H., Teng, D., Mao, R., Hao, Y., Yang, N., Wang, X., & Wang, J. (2020). Dual Antibacterial Activities and Biofilm Eradication of a Marine Peptide-N6NH2 and Its Analogs against Multidrug-Resistant Aeromonas veronii. International journal of molecular sciences, 21(24), 9637.

[7] Moradi, F., Hadi, N., & Bazargani, A. (2020). Evaluation of quorum-sensing inhibitory effects of extracts of three traditional medicine plants with known antibacterial properties. New microbes and new infections, 38, 100769.

[8] Tanveer, T., Ali, S., Ali, N. M., Farooq, M. A., Summer, M., Hassan, A., Ali, F., Irfan, M., Kanwal, L., Shahzad, H., & Islam, R. (2024). Evaluating the Effect of pH, Temperature and Concentration on Antioxidant and Antibacterial Potential of Spectroscopically, Spectrophotometrically and Microscopically Characterized Mentha Spicata Capped Silver Nanoparticles. Journal of fluorescence, 34(3), 1253–1267.

  1. In this study, the authors chose ten virulence-related genes—Aha, exu, lip, ast, alt, act, LuxS, aerA, hlyA, and ascV—for amplifying Aeromonas veronii using conventional PCR with template DNA from the WH10 strain. What was the rationale behind selecting these specific genes to assess WH10's virulence? Is there empirical evidence supporting their relevance in determining bacterial virulence?

Response: Many thanks for your concerns. In this study, we selected ten virulence-related genes (Aha, exu, lip, ast, alt, act, LuxS, aerA, hlyA, and ascV) to assess the virulence of the WH10 strain of Aeromonas veronii. These genes were specifically chosen due to their known association with virulence factors in Aeromonas veronii. For example, cytotoxic enterotoxins encoded by act, alt, and ast are implicated in causing bloody or non-bloody diarrhea, serving as indicators of potential cytotoxic enterotoxicity of Aeromonas spp. [1]. Hemolysin, encoded by hlyA, exerts cytotoxic and lytic effects on red blood cells, aiding bacterial invasion into host tissues [2]. The membrane porin encoded by the Aha gene is crucial for ion transport, adhesion, and the pathogenicity of virulent Aeromonas spp. [3-4].

The extracellular degradation lipase encoded by the lip gene facilitates bacterial evasion of host immune defenses and infection establishment [5]. LuxS is involved in the synthesis of the quorum sensing signal molecule AI-2, a marker gene linked to bacterial virulence and biofilm production [1, 6-10]. The exu gene may encode proteins essential for physiological functions that support the survival and infection capabilities of Aeromonas veronii within the host [11]. Lastly, the ascV gene may play a role in regulating virulence factor expression or be involved in the bacterial secretion system, thereby affecting pathogenicity [12].

In summary, we selected these genes to evaluate the virulence of WH10 based on their known associations with Aeromonas veronii virulence factors. The following literature supports their relevance in determining bacterial virulence.

[1] Chandrarathna, H. P. S. U., Nikapitiya, C., Dananjaya, S. H. S., Wijerathne, C. U. B., Wimalasena, S. H. M. P., Kwun, H. J., Heo, G. J., Lee, J., & De Zoysa, M. (2018). Outcome of co-infection with opportunistic and multidrug resistant Aeromonas hydrophila and A. veronii in zebrafish: Identification, characterization, pathogenicity and immune responses. Fish & shellfish immunology, 80, 573–581.

[2] Kasai, H., Watanabe, K., Gasteiger, E., Bairoch, A., Isono, K., Yamamoto, S., & Harayama, S. (1998). Construction of the gyrB Database for the Identification and Classification of Bacteria. Genome informatics. Workshop on Genome Informatics, 9, 13–21.

[3] Li, T., Raza, S. H. A., Yang, B., Sun, Y., Wang, G., Sun, W., Qian, A., Wang, C., Kang, Y., & Shan, X. (2020). Aeromonas veronii Infection in Commercial Freshwater Fish: A Potential Threat to Public Health. Animals: an open access journal from MDPI, 10(4), 608.

[4] Song, H. C., Kang, Y. H., Zhang, D. X., Chen, L., Qian, A. D., Shan, X. F., & Li, Y. (2019). Great effect of porin(aha) in bacterial adhesion and virulence regulation in Aeromonas veronii. Microbial pathogenesis, 126, 269–278.

[5] Pemberton, J. M., Kidd, S. P., & Schmidt, R. (1997). Secreted enzymes of Aeromonas. FEMS microbiology letters, 152(1), 1–10.

[6] Chen, X., Schauder, S., Potier, N., Van Dorsselaer, A., Pelczer, I., Bassler, B. L., & Hughson, F. M. (2002). Structural identification of a bacterial quorum-sensing signal containing boron. Nature, 415(6871), 545–549.

[7] Chung, W. O., Park, Y., Lamont, R. J., McNab, R., Barbieri, B., & Demuth, D. R. (2001). Signaling system in Porphyromonas gingivalis based on a LuxS protein. Journal of bacteriology, 183(13), 3903–3909.

[8] Frias, J., Olle, E., & Alsina, M. (2001). Periodontal pathogens produce quorum sensing signal molecules. Infection and immunity, 69(5), 3431–3434.

[9] Azakami, H., Teramura, I., Matsunaga, T., Akimichi, H., Noiri, Y., Ebisu, S., & Kato, A. (2006). Characterization of autoinducer 2 signal in Eikenella corrodens and its role in biofilm formation. Journal of bioscience and bioengineering, 102(2), 110–117.

[10] Learman, D. R., Yi, H., Brown, S. D., Martin, S. L., Geesey, G. G., Stevens, A. M., & Hochella, M. F., Jr (2009). Involvement of Shewanella oneidensis MR-1 LuxS in biofilm development and sulfur metabolism. Applied and environmental microbiology, 75(5), 1301–1307.

[11] Nawaz, M., Khan, S. A., Khan, A. A., Sung, K., Tran, Q., Kerdahi, K., & Steele, R. (2010). Detection and characterization of virulence genes and integrons in Aeromonas veronii isolated from catfish. Food microbiology,27(3), 327–331.

  1. Furthermore, it is important to ensure uniformity in formatting the names of virulence genes. Specifically, some genes are italicized while others are written in regular font. Please ensure consistent and correct formatting of all gene names according to scientific conventions.

Response: Thank you very much for pointing out the issue. We will carefully review the formatting of all gene names in the article to ensure they are consistently presented in bold as per scientific conventions. Any inconsistencies will be corrected to maintain uniformity throughout the text.

  1. For the experiments of bacterial isolation and identification, it is recommended to provide more background information about sample collection and processing, such as the number of samples collected, sources and storage conditions, so as to enhance the repeatability of the experiments.

Response: Many thanks. For the experiments involving bacterial isolation and identification, we will provide more detailed information on sample collection and processing:

A total of 10 diseased bronze gudgeon and 540 healthy bronze gudgeons were collected for the experiments. The healthy fish were divided into six groups of 30 fish each, with three replicates for each experiment.

Both diseased and healthy fish were sourced from the Institute of Water Engineering Ecology, Chinese Academy of Sciences, Ministry of Water Resources, Wuhan, Hubei Province. Diseased fish were kept on ice immediately after collection and transported to the laboratory for processing as quickly as possible. Healthy fish were stored in oxygenated bags and transported to the laboratory promptly. Upon arrival, they were placed in a pre-prepared breeding tank with an oxygenation and circulating water system.

To acclimatize the healthy fish to the breeding tank environment and minimize stress, the oxygenated bags containing the fish were placed in the tank for two hours. After this acclimation period, the fish were released into the breeding tank for the experiments.

  1. In the analysis of blood parameters, it is essential for the authors to give a detailed description of the time and method of blood sample collection. Additionally, they should outline measures taken to ensure the stability and accuracy of the samples throughout the process.

Response: Thank you for the suggestions. For blood sample collection, we will provide more detailed information as follows:

During the experiment, blood samples were collected from three diseased and three healthy bronze gudgeon using disposable sterile syringes. The blood was immediately used to prepare blood smears on slides, with three smears taken for each fish to ensure the accuracy and reliability of the results. Reichs-Giemsa staining was performed to classify and count 100 white blood cells on each smear.

To quantify serum biochemical indicators, 1.5 mL of fresh blood was collected from both diseased and healthy fish. The blood was stored at 4°C overnight and then centrifuged at 4000 g for 10 minutes. During this process, samples were carefully handled, and the supernatant was transferred to new test tubes for further analysis. The experiment was repeated three times, following strict procedural guidelines to minimize errors.

  1. In conjunction with the antibiotic sensitivity results of Aeromonas veronii in this study, the discussion should explore potential causes and influencing factors by comparing and analyzing findings from related studies.

Response: We gratefully appreciate your valuable comments. Testing the antibiotic response of WH10 demonstrated its sensitivity to compound sulfamethoxazoles, cefotaxime, doxycycline, and sulfamethoxazole. It exhibited moderate sensitivity to gentamicin, amikacin, neomycin, florfenicol, and enrofloxacin, while displaying resistance to ciprofloxacin. These results differ from previously reported sensitivities for Aeromonas veronii [1, 2, 3]. This variability in antibiotic sensitivity might be attributed to differences in the strains' origins or their distinct antibiotic resistance profiles.

  • Xu, X., Fu, H., Wan, G., Huang, J., Zhou, Z., Rao, Y., Liu, L., & Wen, C. (2022). Prevalence and genetic diversity of Aeromonas veroniiisolated from aquaculture systems in the Poyang Lake area, China. Frontiers in microbiology, 13, 1042007.
  • Dos Santos, S. B., Alarcon, M. F., Ballaben, A. S., Harakava, R., Galetti, R., Guimarães, M. C., Natori, M. M., Takahashi, L. S., Ildefonso, R., & Rozas-Serri, M. (2023). First Report of Aeromonas veroniias an Emerging Bacterial Pathogen of Farmed Nile Tilapia (Oreochromis niloticus) in Brazil. Pathogens (Basel, Switzerland), 12(8), 1020.
  • Legario, F. S., Choresca, C. H., Jr, Grace, K., Turnbull, J. F., & Crumlish, M. (2023). Identification and characterization of motile Aeromonas spp. isolated from farmed Nile tilapia (Oreochromis niloticus) in the Philippines. Journal of applied microbiology, 134(12), lxad279.
  1. Figure 1: It is recommended to adjust the color of the indicating arrow on clinically diseased fish to improve visibility against the background. For instance, in Figure 1B and D, the arrow color blends with the background, making it challenging to discern clearly.

Response: Many Thanks. We will adjust the color of the triangular arrowheads (navy blue triangular arrowheads) indicating clinically diseased fish in Figure 1 to ensure it contrasts clearly with the background. This adjustment will enhance visibility, allowing readers to identify it more easily.

Reviewer 3 Report

Comments and Suggestions for Authors

The manuscript is of scientific interest and generally in the scope of the journal, being devoted to the study of Aeromonas veronii, a pathogen of many animals. The merit of the work is its comprehensiveness; many aspects of the biology, physiology, biochemistry, pathogenesis and antibiotic susceptibility of this bacteria were studied, and the species identification was carried out based on an integrative approach. Overall, the article is well structured, the results are clearly presented, and the discussion is sufficient. All this makes this manuscript worthy of publication in the journal.

 I have only one substantial question for the authors.

 1.    In the 'Materials and Methods' section of the description of the experiment (lines 125-128), it is stated that the experimental gudgeons were "… challenged via intraperitoneal (IP) injection with 0.3 mL of bacterial supernatant at titers of 1× 104.0, 1× 105.0, 1× 106.0, 1× 107.0 and 1× 108.0 colony forming units (CFU)/mL, respectively”.

However, in the results presented in section 3.10 Pathogenicity (lines 296-299) and in Figure 6, other concentrations are given: 2× 104.0, 2× 105.0, 2× 106.0, 2× 107.0 and 2× 108.0.

How were these concentrations obtained? since 0.3 mL × 1× 104.0 = 3× 103.0, etc.

It is not clear what is meant by "individuals in the experimental group receiving 2 × 108 CFU/mL fish weight ". If the doses were normalised to fish weight, this should be added in the Methods.

 This disagreement needs to be clarified, either it is an error or information needs to be added to the Methods to make it clear how the concentrations were obtained as in the Results.

 There are also some minor remarks:

 2. Lines 38-39: “The bronze gudgeon (Coreius heterodon) is an indigenous fish species in China, classified under the subfamily Gobioninae within the order Cypriniformes.”

Hereinafter, when taxa are mentioned for the first time, the authors and the year of description of the taxon should be given. For example, Coreius heterodon (Bleeker, 1864) and Aeromonas veronii Hickman-Brenner, MacDonald, Steigerwalt, Fanning, Brenner & Farmer, 1987, etc.

3. The headings of the second and third columns in Table 3 are identical. It appears that the third column is not in mm. Correct this.

4. The caption for Figure 5 (lines 274-281) repeats the text from lines 264-278. Similarly, the continuation of the figure caption repeats the text from lines 284-290. These repetitions should be removed from the figure captions and left only in the text.

Author Response

Reviewer #3:

The manuscript is of scientific interest and generally in the scope of the journal, being devoted to the study of Aeromonas veronii, a pathogen of many animals. The merit of the work is its comprehensiveness; many aspects of the biology, physiology, biochemistry, pathogenesis and antibiotic susceptibility of this bacteria were studied, and the species identification was carried out based on an integrative approach. Overall, the article is well structured, the results are clearly presented, and the discussion is sufficient. All this makes this manuscript worthy of publication in the journal.

 I have only one substantial question for the authors.

  1. In the 'Materials and Methods' section of the description of the experiment (lines 125-128), it is stated that the experimental gudgeons were "… challenged via intraperitoneal (IP) injection with 0.3 mL of bacterial supernatant at titers of 1× 104.0, 1× 105.0, 1× 106.0, 1× 107.0 and 1× 108.0 colony forming units (CFU)/mL, respectively”.

However, in the results presented in section 3.10 Pathogenicity (lines 296-299) and in Figure 6, other concentrations are given: 2× 104.0, 2× 105.0, 2× 106.0, 2× 107.0 and 2× 108.0.

How were these concentrations obtained? since 0.3 mL × 1× 104.0 = 3× 103.0, etc.

It is not clear what is meant by "individuals in the experimental group receiving 2 × 108 CFU/mL fish weight ". If the doses were normalised to fish weight, this should be added in the Methods.

This disagreement needs to be clarified, either it is an error or information needs to be added to the Methods to make it clear how the concentrations were obtained as in the Results.

Response: We apologize for any confusion caused to the reviewers. In this experiment, the selected concentration of infectious bacteria was 2× 104.0, 2× 105.0, 2× 106.0, 2× 107.0 and 2× 108.0 CFU/mL; The reference to 1× 104.0, 1× 105.0, 1× 106.0, 1× 107.0 and 1× 108.0 CFU/mL was a typographical error. We will correct this in the corresponding sections of the article. We appreciate the reviewer for pointing out this mistake.  

There are also some minor remarks:

  1. Lines 38-39: “The bronze gudgeon (Coreius heterodon) is an indigenous fish species in China, classified under the subfamily Gobioninae within the order Cypriniformes.”

Hereinafter, when taxa are mentioned for the first time, the authors and the year of description of the taxon should be given. For example, Coreius heterodon (Bleeker, 1864) and Aeromonas veronii Hickman-Brenner, MacDonald, Steigerwalt, Fanning, Brenner & Farmer, 1987, etc.

Response: Thank you for your helpful suggestions. We have included the author's name and the year when the taxa are first mentioned following your suggestions.

  1. The headings of the second and third columns in Table 3 are identical. It appears that the third column is not in mm. Correct this.

Response: Thank you for pointing out this problem in the manuscript. We have noticed that the second and third column headings in Table 3 are identical, and we will correct this error. Upon verification, we confirm that the units in the third column are indeed millimeters.

4.The caption for Figure 5 (lines 274-281) repeats the text from lines 264-278. Similarly, the continuation of the figure caption repeats the text from lines 284-290. These repetitions should be removed from the figure captions and left only in the text.

Response: Thank you for your helpful suggestions. We will remove any redundant text from the Figure 5 header to ensure it is concise and clear while maintaining the accuracy of the caption information.

Round 2

Reviewer 1 Report

Comments and Suggestions for Authors

Paper can be accepted for publication

Author Response

Dear Reviewer:

Thank you so much for your thoughtful comments and for taking the time to review our manuscript. We sincerely appreciate your valuable feedback.

Best regards, 

Wenzhi Liu